# Association of SO_2_-Generating Pads before Packaging and during Cold Storage to Extend the Conservation of ‘Italia’ Table Grapes

**DOI:** 10.3390/plants13192827

**Published:** 2024-10-09

**Authors:** Maíra T. Higuchi, Aline C. de Aguiar, Nathalia R. Leles, Viviani V. Marques, Leandro S. A. Gonçalves, Fábio Yamashita, Khamis Youssef, Sergio R. Roberto

**Affiliations:** 1Agricultural Research Center, Agronomy Department, State University of Londrina, Celso Garcia Cid Road, km 380, Londrina 86057-970, Brazil; maira.tiaki.higuchi@uel.br (M.T.H.); aguiar.alinec@gmail.com (A.C.d.A.); nathalia.leles@uel.br (N.R.L.); vivianimarques@yahoo.com.br (V.V.M.); leandrosag@uel.br (L.S.A.G.); fabioy@uel.br (F.Y.); 2Agricultural Research Center, Plant Pathology Research Institute, 9 Gamaa St., Giza 12619, Egypt; youssefeladawy@yahoo.com

**Keywords:** *Botrytis cinerea* Pers, sulfur dioxide, plastic liners, gray mold, *Vitis vinifera* L.

## Abstract

The SO_2_-generating pads contain different concentrations of sodium metabisulfite, which absorbs water from the grapes’ transpiration, releasing SO_2_ gas, and there are slow-(SlowSO_2_) and dual (DualSO_2_)-releasing pads (fast release in the first 48 h and slow for up to 60 days). The ultra-fast SO_2_-generating pad (FieldSO_2_) releases the SO_2_ quickly for up to 6 h, and it was designed to be used soon after the harvest and until the grapes’ packaging. The goal was to study the effect of FieldSO_2_ associated with SlowSO_2_ and DualSO_2_ pads on gray mold incidence and physicochemical and appearance characteristics of ‘Italia’ table grapes. Grapes were harvested from a commercial vineyard in Parana, Brazil, in 2020 and 2021, and packaged in cardboard boxes, and the treatments were as follows: control (without SO_2_-generating pads); FieldSO_2_ + SlowSO_2_; and FieldSO_2_ + DualSO_2_. After 30, 45, 60, 75, and 90 days of cold storage (1 ± 1 °C), the grapes were assessed for gray mold incidence, mass loss, shattered berries, stem browning, and filamentous fungi on the surface. The use of FieldSO_2_ associated with SO_2_-generating pads is effective in controlling gray mold on ‘Italia’ table grapes, especially the treatment FieldSO_2_ + DualSO_2_, which provides the lowest incidence of the disease up to 90 days of cold storage, while the combination with SlowSO_2_ results in intermediate efficacy. Treatments combining these SO_2_-generating pads extend the postharvest shelf life of ‘Italia’ grapes, with few shattered berries, low mass loss and freshness of the rachis without impairing the bunch’s appearance.

## 1. Introduction

‘Italia’ table grapes (*Vitis vinifera* L.), a popular cultivar in global markets, are prized for their large size, crisp texture, and sweet flavor. This variety is favored for its exceptional taste and appearance and its versatility for fresh consumption. This cultivar plays a significant role in the agricultural economies of many countries, particularly in regions like Southern Europe and South America, where the climate favors their cultivation. The cultivar’s resistance to transportation and its good shelf life make it an excellent choice for export, contributing to the global table grape industry [1,2].

Among the attributes that influence the acceptance of table grapes, appearance is fundamental, and it is related to the color of the rachis and berries, the shape and size of the bunch, and the absence of injuries and decay, among others [3].

After harvest, table grapes can suffer injuries from handling, water loss, and pathogen attacks [4], reducing their quality and harming their commercialization, mainly for export [5]. Therefore, it is essential to maintain their sensorial characteristics, mainly appearance, since the grapes are subjected to long periods of cold storage until arrival at their end destination [6].

Some cultivars of table grapes can be stored for prolonged periods using appropriate packaging associated with cold storage throughout the entire commercialization chain, especially when transport is carried out over long distances. The cold chain breaking and the high relative humidity (RH) required to avoid dehydration favor the development of pathogens from latent infections, which can result in grapes with rot symptoms, impairing their commercialization [7].

The main postharvest disease of table grapes is gray mold, caused by the fungus *Botrytis cinerea* Pers. Conditions of high RH and room temperature are favorable for the development of this pathogen [8], but it can develop at low temperatures (±0.5 °C), spreading throughout all berries [9], making it essential to adopt measures for its control [10,11].

Pads that generate sulfur dioxide gas (SO_2_), which inhibits fungal growth, stand out in environments with high humidity [12]. The use of SO_2_-generating pads inside the cardboard boxes with grapes has demonstrated excellent performance in preventing the development of gray mold and prolonging the shelf life of the fruits during cold storage [5,13,14,15]. The SO_2_ also influences the fruit’s physiological processes, maintaining the freshness of the rachis due to its antioxidant action [7].

The SO_2_-generating pads can contain different concentrations of the active ingredient (AI) sodium metabisulfite (Na_2_S_2_O_5_), which absorbs water from the grapes’ transpiration, releasing SO_2_ gas. Commercially, there are two models of pads, slow-(SlowSO_2_) and dual (DualSO_2_)-releasing pads (fast release in the first 48 h and slow for up to 60 days) [6], and the use of each of them depends on several factors, including the sensitivity of each grape variety and climatic conditions, among others. In addition to being efficient in controlling rot, this technique is easy to use, affordable, and has a low health risk compared to fungicides applied in the field [16].

Whether housed in clamshells or not, the grape bunches are packaged in perforated plastic liners with different ventilation areas. The objective is to reduce the mass loss caused by fruit dehydration and enable better efficiency of the SO_2_-generating pads [5].

The vented plastic clamshells are innovative and successful packaging for grapes [7]. Storing the bunches individually in the cardboard box avoids their physical contact with the external environment, improves the attractiveness and practicality for the consumer, and preserves the bunches’ integrity [17,18]. In this packaging, losses caused by shattered berries no longer exist since the loose berries are marketable as they remain inside the bowl. Furthermore, the bunches do not have physical contact with each other in the cardboard box, which prevents possible gray mold spread.

Although all of these measures depend on the grape cultivar and pre-harvest climatic conditions, gray mold can develop during long periods of cold storage. The ultra-fast SO_2_-generating pad (FieldSO_2_) was developed to be used between the harvest and the grapes’ packaging. Due to their high permeability, these pads release SO_2_ quickly for up to 6 h. Thus, the harvested grapes are immediately placed in harvesting boxes with a 20 kg capacity lined with perforated plastic liners. When the harvest box is filled, the FieldSO_2_ pad is placed on top of the grapes, and the liner is sealed until packaged in the packing house. Due to the high humidity inside the perforated plastic liner, the grapes are treated with SO_2_ gas for 4–6 h before packaging [19,20]. So, the FieldSO_2_ pad has an eradicating effect on B. cinerea spores found on the surface of the berries, reducing the disease development during cold storage [21].

In previous trials, we found that using FieldSO_2_ associated with SO_2_-generating pads during cold storage results in a lower incidence of gray mold after 45 days of storage [22]. However, to our knowledge, the effectiveness of combinations of these pads to extend the table grapes conservation for longer periods is still unknown. Therefore, this work aimed to assess gray mold incidence, mass loss, stem browning, and shattered berries of ‘Italia’ table grapes subjected to the FieldSO_2_ pad associated with SlowSO_2_ and DualSO_2_ pads up to 60 days of cold storage.

## 2. Results

### 2.1. Season of 2020

In the 2020 season, significant differences were observed among the treatments regarding the incidence of gray mold, mass loss, shattered berries, and stem browning of the ‘Italia’ table grape. The development of the incidence of gray mold was exponential, while the other assessed characteristics of the grapes were best fitted to the linear model for all treatments assessed over the 90 days of cold storage (Figure 1, Appendix A).

The grapes treated with FieldSO_2_ and SO_2_-generating pads during cold storage had a lower incidence of gray mold than the control treatment in all periods assessed. After 60 days of cold storage, the FieldSO_2_ + DualSO_2_ treatment had the highest efficiency in controlling gray mold in grapes, differing from those subjected to the FieldSO_2_ + SlowSO_2_ treatment. After 90 days of cold storage, the bunches of the control treatment had abundant sporulation of *B. cinerea*, covering part of the surface of the berries (Figure 2).

These results were confirmed by quantifying the filamentous fungi population on the grape berries’ surface before and after the bunches were subjected to FieldSO_2_ and throughout cold storage (Figure 3). Filamentous fungi were not detected on berry skins after the bunches had been subjected to FieldSO_2_ and FieldSO_2_ + DualSO_2_ in all periods assessed due to the eradicating effect of fungal spores with these treatments, in addition to preventing the development of latent fungal structures in the berries. The highest mean was found in bunches from the control treatment, followed by the FieldSO_2_ + SlowSO_2_, which had an intermediate eradicating effect. It was found that the presence of the fast-release phase of SO_2_ in the first 48 h of cold storage of the DualSO_2_ pad resulted in a significant increase in the effectiveness of disease control.

The mass loss of the bunches of grapes had no difference among the treatments up to 45 days of cold storage, and after 90 days, the highest mass loss was observed in bunches from the control treatment, while those subjected to the FieldSO_2_ + SlowSO_2_ had the lowest mean. After 45 days of cold storage, the grapes subjected to the FieldSO_2_ + DualSO_2_ had the least stem browning, which means fresher stems, while those from the control treatment had the most browning, which means high stem oxidation. The differences in shattered berries from the bunches of grapes among the treatments occurred only at 90 days of cold storage, in which grapes subjected to FieldSO_2_ + SlowSO_2_ had the lowest mean of loose berries, while the highest mean was found in the bunches subjected to the FieldSO_2_ + DualSO_2_ treatment.

No significant differences were observed among the grapes from all treatments regarding the chemical properties of the berries, such as soluble solids, titratable acidity, and maturity index throughout the cold storage period (Appendix A), which indicates that the treatments did not impact these postharvest characteristics of the grapes.

### 2.2. Season of 2021

In the 2021 season, significant differences were found among the treatments for the incidence of gray mold and shattered berries of the ‘Italia’ table grape. As in the previous season, it was also found that the development of the incidence of gray mold was exponential, while for the other characteristics of the grapes, the development best fitted linear regression for all treatments assessed over the 90 days of cold storage (Figure 4, Appendix A).

Like in the 2020 season, grapes subjected to the FieldSO_2_ + SlowSO_2_ and FieldSO_2_ + DualSO_2_ treatments had a lower incidence of gray mold than those of the control treatment in all periods assessed. After 60 days of cold storage, the grapes subjected to the FieldSO_2_ + DualSO_2_ treatment began to stand out in terms of higher efficacy in controlling gray mold, differing from the bunches subjected to the FieldSO_2_ + SlowSO_2_ treatment, which, in turn, also differed from the control. These results are supported by assessing the population of filamentous fungi on the surface of the berry skins before and after the bunches had been subjected to FieldSO_2_ treatment, as well as throughout the cold storage period (Figure 5). In this assessment, these fungi were not detected in the bunches after being subjected to the FieldSO_2_ and FieldSO_2_ + DualSO_2_ treatments in all periods assessed, confirming their higher efficiency in eradicating fungal spores from the surface of the berries before packaging.

The mass loss of the bunches of grapes had no difference among the treatments up to 90 days of cold storage. After 60 days of cold storage, there were no differences among the treatments regarding stem browning, and after 75 days, the grapes subjected to the FieldSO_2_ + DualSO_2_ showed the least stem browning, which means fresher stems, while those from the control treatment had the most browning. The shattered berries from the bunches of grapes subjected to FieldSO_2_ + DualSO_2_ had the lowest mean of loose berries, and those from the control treatment had the highest mean during the entire storage time.

As in the previous season, there were no significant differences among the treatments for the postharvest chemical properties of the berries throughout the cold storage period (Appendix A), which indicates that any of the treatments did not impact the properties of the grapes.

## 3. Discussion

In both seasons (2020 and 2021), the FieldSO_2_ + SlowSO_2_ and FieldSO_2_ + DualSO_2_ treatments effectively controlled gray mold in ‘Italia’ grapes during 90 days of cold storage. Treating the bunches immediately after harvesting with FieldSO_2_ for about 5 h, the B. cinerea spores on the surface of the berry skins were eradicated with high efficacy, and the berries were disinfected before packaging.

The eradicating action due to the FieldSO_2_ treatment, associated with a low and continuous SO_2_ emission by the SlowSO_2_ pad during cold storage, maintained the low incidence of gray mold in grapes after 90 days at 1 °C and 70–80% RH, and the association of FieldSO_2_ + DualSO_2_ treatments had the lowest incidence of the disease in both seasons. The higher amount of gas released in the first 48 h of cold storage possibly contributed to eradicating fungi spores that survived after the FieldSO_2_ pad treatment. The combination of FieldSO_2_ fast phase with slow and continuous gas released by the DualSO_2_ pad had the best performance in controlling gray mold, as reported by other authors [6,23,24]. It is also important to add that other commercial technologies, such as the use of SO_2_-gassing chambers, can also be used to preserve table grapes during cold storage with a similar yield. In this technology, the SO₂ gas is introduced into the chamber, often in controlled doses to achieve the desired concentration without exceeding regulatory limits. Grapes are usually exposed to the gas for a specific period, often a few hours, depending on the concentration, and proper gassing can significantly prolong the storage life of table grapes, allowing them to be transported over long distances without spoilage.

The performance of the FieldSO_2_ treatment, as well as the DualSO_2_, used alone and in combination, was assessed by to control gray mold on ‘BRS Nubia’ table grapes up to 45 days of cold storage, and differences were found among the treatments, but the authors reported an absence of gray mold symptoms when these two treatments were used in combination [19]. Other authors reported a lower incidence of gray mold in ‘Benitaka’ table grapes subjected to FieldSO_2_ associated with a permeable bio-based liner, extending the shelf life of the grapes for up to 45 days in cold storage [25].

In this work, the FieldSO_2_ + DualSO_2_ treatment was not able to completely prevent the emergence of gray mold symptoms in cold storage due to the endophytic behavior of the fungus, which can survive latently in the host [26], and the insufficient residual effect of SO_2_ in the berries to suppress its development for more extended storage [27]. The AI concentration in pads must guarantee a regular supply of SO_2_ until the end of storage because the gas does not reach the berry epicarp; then, the grapes must be continuously exposed to the gas, thus preventing the fungal mycelium growth [28,29].

However, the higher AI concentration of the FieldSO_2_ and the DualSO_2_ pads of this treatment (1.4 g + 5.0 g of AI, respectively) was appropriate, as it provided a satisfactory control of gray mold after 90 days of cold storage, with a very low incidence of the disease, with no damage to the bunches, such as bleaching and berry cracking, early stem browning or unpleasant taste being detected [30,31].

Based on these results, the SO_2_ released by the generating pads remained circulating in the packaging throughout the cold storage period, and the vented plastic clamshells contributed to keeping it in contact with the grapes for longer, extending its effectiveness. This packaging, despite being considered a physical barrier to the uniform circulation of SO_2_, maintained a suitable environment by retaining the gas around the bunches [5,14,30,32]. The perforated plastic liner enhanced the SO_2_-generating pad effect during cold storage, retaining the gas in the cardboard boxes and allowing for circulation inside the packaging. The liner also reduced the mass loss of the grapes [33,34] because it is a barrier to water vapor transmission, and the lower the transmission rate, the higher the relative humidity inside the packaging, thus reducing the transpiration and, consequently, bunch dehydration [35,36]. Therefore, using a perforated plastic liner, combined or not with the SO_2_-generating pad, is necessary to reduce water loss during postharvest handling of table grapes [37]. The lower incidence of gray mold using an SO_2_-generating pad also contributes to minimizing the mass loss because the parasitism process triggered by B. cinerea can supposedly increase the respiratory rate of bunches [19].

Postharvest management is essential to maintaining the quality of fresh fruits and vegetables because consumers expect the product to be fresh and preserve its functional characteristics [38]. In general, treatments with SO_2_ and low storage temperatures kill most arthropod pests and several microorganisms (e.g., spores of *Botrytis* on the surface of the berry skin). Proper packaging material ensures that the gas has a good diffusion and reaches every fruit with adequate concentrations [39] because, depending on the grape’s sensitivity to the SO_2_, it can cause damage to the berries, like bleaching.

The proper use of SO_2_-generating pads allows for table grape storage for several weeks or even months, and during shipments exceeding 10 days, SO_2_-generating pads ensure a constant concentration of the SO_2_ gas according to the amount of the AI. However, storage and transport time is variable; for instance, transport time by ship from South Africa, Chile, Brazil, or India to the Netherlands averages three weeks [26].

The grapes’ mass loss during cold storage was linear (Figure 1B), and the mass loss rates were similar for all treatments (control, FieldSO_2_ + SlowSO_2_, and FieldSO_2_ + DualSO_2_), around 0.11% mass loss day^−1^ in the 2020 season, and 0.10% mass loss day^−1^ in the 2021 season. During cold storage, despite the packaging and high RH, the grapes lose mass due to the transpiration and respiration of the berries [40].

After 90 days of storage, the mass loss ranged from 6.82% to 8.33% in the 2020 season and 7.07% to 7.19% in the 2021 season. The berries did not have symptoms of shriveling because the ‘Italia’ grapes are large and firm with a turgid appearance [2] and less susceptible to shrinkage due to mass loss. When the mass loss is severe, it leads to dehydration of the stem and, consequently, darkening, followed by shattered berries. However, the mass loss after 90 days of cold storage can be considered acceptable, resulting in a low percentage of shattered berries and stems with slight browning, with no impact on the appearance of the bunches.

The SO_2_ gas, besides inhibiting the development of microorganisms, has an antioxidant action, influencing the postharvest physiology of table grapes, keeping the stem green and fresh and the berries turgid for longer periods [41,42], due to the inhibitory action of SO_2_ on the catalytic mechanism of some enzymes that favor the respiratory process [43]. Studies reported that the stems of ‘Shine Muscat’ table grapes treated with SO_2_ had a higher accumulation of phenolic compounds by increasing the transcript levels of *VvLAR* and *Vv3GT* genes involved in the flavonoid pathway [44]. The authors also reported that SO_2_ treatment reduced the separation degree of plasma membrane from the cell wall and maintained the homogeneity of cytoplasm and integrity of organelles in rachis, indicating that SO_2_ alleviates the damage and degradation of the cellular structure in the rachis.

However, excess SO_2_ can cause damage to the cell wall structure of table grapes, leading to premature abscission of the berries during storage [45], which was not observed in this work. The percentage of shattered berries was low for all treatments, and berry detachment was probably due to abscission, a natural postharvest physiological phenomenon [46]. Besides that, the ‘Italia’ is a seeded grape, with berries firmly attached to the pedicels, and it presents great resistance to shattered berries [2].

The use of SO_2_-generating pads during cold storage is a strategy for controlling gray mold in table grapes, mainly by using the DualSO_2_ pad [6,47,48]. Several reports are available about its effectiveness on table grapes grown under a two-cropping-a-year system, with cold storage periods varying from 45 to 50 days [5,13,14,15]. However, when considering long storage periods, such as those evaluated in this work, the association with the FieldSO_2_ treatment extended the grapes’ shelf life to 90 days by controlling gray mold development and maintaining their physicochemical characteristics.

We have demonstrated in this work that the FieldSO_2_ + DualSO_2_ treatment is an effective strategy to extend the postharvest shelf life of the ‘Italia’ table grape up to 90 days at 1 °C without symptoms of gray mold disease. This is a significant achievement, especially for grapes destined for export from Brazil to the main destinations, such as Europe and Asia, because logistics can take several weeks from harvesting to the supermarket shelves. The technology also offers a solution to the issue of production peaks in the market. The Brazilian domestic market for table grapes could benefit from this technology because several supermarket chains demand that the grapes be packaged similarly to those exported, aiming for their conservation in cold storage in distribution centers and later in the retail units. Prolonged storage of table grapes using the FieldSO_2_ + DualSO_2_ can be a strategy for grape growers to sell their production during periods of lower supply of grapes on the market, avoiding production peaks, expanding the offer period, and thus obtaining more advantageous prices.

## 4. Materials and Methods

### 4.1. Location

Fully ripened bunches of ‘Italia’ grapes (*Vitis vinifera* L.) used in this work were harvested from a commercial vineyard located in the municipality of Cambira, Paraná, Brazil (23°34′58″ S, 51°34′40″ W, elevation of 1017 m). Grapevines were grafted onto the 11-year-old ‘IAC-766 Campinas’ rootstock, trained in an overhead trellis system and covered by black plastic mesh of 18% shading; the region’s climate, according to the classification proposed by Köppen, is subtropical Cfa, with an average annual precipitation of 1633.5 mm, an average minimum temperature of 18 °C and a maximum of 22 °C [49]. The bunches of the ‘Italia’ grape were harvested in two consecutive seasons, 2020 and 2021, when the grapes reached the ideal ripening stage, around 14 ºBrix. The area was selected because it has a recurrent record of gray mold [5,31,32].

### 4.2. Treatments and Experimental Design

Based on our previous results [22], the treatments were associated with different packaging systems to extend the cold conservation of ‘Italia’ grapes up to 90 days, as follows: (a) control (without SO_2_-generating pads); (b) field ultra-fast SO_2_-generating pad before packaging associated with slow-SO_2_-releasing pad during cold storage (FieldSO_2_ + SlowSO_2_); and (c) field ultra-fast SO_2_-generating pad before packaging associated with dual-SO_2_-releasing pad during cold storage (FieldSO_2_ + DualSO_2_).

The FieldSO_2_ (1.4 g of active ingredient-AI, sodium metabisulphite), SlowSO_2_ (4 g of AI), and the DualSO_2_ pads (5 g of AI) (Grape Guard Uvas Quality^®^), as well the perforated plastic liners (0.3% of ventilation area) were supplied by IMAL, SpA, Santiago, Chile. The size of all tested SO_2_-generating pads was 46 × 26 cm. The FieldSO_2_ pads containing 1.6 g of AI were designed to be used in the field right after the grapes are harvested by releasing the SO_2_ gas in a fast way during 4–6 h, killing the fungi spores from berry skins before packaging, while the SlowSO_2_ and the DualSO_2_ pads were designed to be used inside the cardboard boxes during cold storage. The SlowSO_2_ pads containing 5 g of AI release the SO_2_ gas in an even way during the storage period, while the DualSO_2_ pads release the SO_2_ gas in a fast way during the first 48 h (1 g of AI) and in an slow and even way (4 g of AI) during the storage period (see other details in Section 4.3). Clamshells containing grapes were packed in cardboard boxes lined with perforated plastic liners for all treatments. No SO_2_-generating pads were used as a control treatment; however, perforated plastic liners with a 0.3% ventilation area (VA) were used to prevent bunch dehydration. A completely randomized design was used as the statistical model with four replicates, and each plot consisted of a corrugated cardboard box containing 10 bunches individually packaged in clamshells.

### 4.3. Bunch Packaging

The bunches subjected to FieldSO_2_ treatments were placed in plastic harvest boxes (capacity of 20 kg), previously lined with a perforated plastic liner with 0.3% VA [22]. Then, a FieldSO_2_ pad was placed on the grapes, and the liners were sealed, leaving the bunches exposed to this condition for 5 h. During this period, between the transportation of the grapes from the field to their packaging in the packing house, the concentration of SO_2_ emitted by FieldSO_2_ was quantified by using a passive dosimeter (Gastec Passive Dosi-Tube, Kanagawa, Japan), and the readings were 160, 410 and 600 ppm·h after 1, 2 and 3 h, respectively.

The eradicating effect of fungal spores of the FieldSO_2_ was determined by quantifying the epiphytic populations of filamentous fungi on the surface of the berries before and after the bunches had been subjected to this treatment [50]. At each evaluation period, before and after treatment with FieldSO_2_, 10 berries were placed in triplicate in 100 mL of sterile distilled water on a rotary shaker at 150 rpm for 30 min. Then, a 0.1 mL suspension was inoculated per plate into a PDA culture medium containing ampicillin and streptomycin sulfate (250 mg L^−1^ of each antibiotic). The inoculated plates were incubated at 24 °C, and after 4 days, the number of colonies expressed as log_10_ CFUberry was counted.

After the FieldSO_2_ treatment, the bunches were cleaned, eliminating the injured berries, standardized to approximately 0.5 kg, and individually packaged into vented plastic clamshells with 0.5 kg capacity and dimensions of 18.5 × 11.5 × 8.5 cm (L, W, H) each, containing 4 ventilation holes with a 0.5 cm diameter in the hinge.

The grape-packaging process was carried out according to the following steps: the corrugated cardboard boxes (38 × 31 × 10 cm) with a storage capacity of 5 plastic clamshells were internally lined with a perforated plastic liner of 0.3% VA. Above the liner, at the bottom of the box, a sheet of unilaminar moisture-absorbing paper (37 × 28 cm) was placed. The clamshells with the bunches subjected to FieldSO_2_ were placed in the cardboards, and an SO_2_-generating pad (SlowSO_2_ or DualSO_2_) was placed above them, according to the treatment. Finally, the plastic liners were sealed with adhesive tape. The cardboard was cold-stored for 90 days at 1.0 ± 1.0 °C and 70–80% relative humidity.

### 4.4. Assessments of Incidence of Gray Mold, Mass Loss, Shattered Berries, Stem Browning and Population of Filamentous Fungi on Berry Surface

At 30, 45, 60, 75, and 90 days of cold storage, the following variables were analyzed: incidence of gray mold, mass loss, shattered berries, stem browning, and quantification of filamentous fungi on the surface of the berries. At 45 and 90 days of cold storage, the chemical properties of the berry juice were also analyzed.

The incidence of gray mold (% of diseased berries) was calculated by Equation (1) [50]: (1)$$\mathrm{Incidence}\left(\%\right)=\frac{\mathrm{number}\;\mathrm{of}\;\mathrm{diseased}\;\mathrm{berries}}{\mathrm{total}\;\mathrm{berries}}\times \;100$$

Stem browning was evaluated by visual assessment [23], assigning the following scores corresponding to the levels of stem or rachis browning: (1) fresh and green, (2) light browning, (3) significant browning, and (4) severe browning.

Bunch mass loss was calculated by weighing the bunches at the initial time of storage and during each evaluation by using Equation (2) [51]:(2)$$\mathrm{Mass}\;\mathrm{loss}\;\left(\%\right)=\frac{\mathrm{initial}\;\mathrm{mass}-\mathrm{mass}\;\mathrm{at}\;\mathrm{examinated}\;\mathrm{date}\;}{\mathrm{initial}\;\mathrm{mass}}\times \;100$$

Shattered berries’ incidence was assessed by counting the loose berries from the bunch inside the clamshells expressed as a percentage [30].

The chemical properties of the berries were assessed by extracting the juice from 10 berries per plot to determine the soluble solids (SSs) and titratable acidity (TA) contents, maturity index (SS/TA), and pH [50].

The epiphytic populations of filamentous fungi on the surface of the berry skins, including *B. cinerea*, *Penicillium* spp., *Aspergillus* spp. and *Rhizopus stolonifer,* were also quantified at 30, 45, 60, 75, and 90 days of cold storage from 10 berries per plot at each assessment period, according to the methodology previously described.

### 4.5. Statistical Analysis

Data on the incidence of gray mold, stem browning, shattered berries, and mass loss were subjected to analysis of variance at 5% significance. The Shapiro–Wilk and Bartlett tests were used to test the assumptions of errors’ normality and variances’ homogeneity, respectively. If significant, a polynomial regression analysis was performed to analyze the development of the assessed characteristics of the ‘Italia’ grape considering the split plot arrangement subdivided in time.

For the incidence of gray mold, the exponential model for this characteristic was used in this work, as disease progress curves tend to have this behavior at the beginning of their development. To quantify filamentous fungi on the surface of berry skins, the analysis of variance and subsequent Tukey’s test (*p* < 0.05) were performed within each assessment period. All analyses were carried out using the R-Studio software [52].

## 5. Conclusions

The use of FieldSO_2_ associated with SO_2_-generating pads is effective in controlling gray mold on ‘Italia’ table grapes, especially the treatment with FieldSO_2_ + DualSO_2_, which provides the lowest incidence of the disease up to 90 days of cold storage, while the combination with SlowSO_2_ results in intermediate efficacy. Treatments combining these SO_2_-generating pads extend the postharvest shelf life of ‘Italia’ grapes, with few shattered berries, low mass loss and freshness of the rachis without damage to the bunch’s appearance.

## Figures and Tables

**Figure 1 plants-13-02827-f001:**
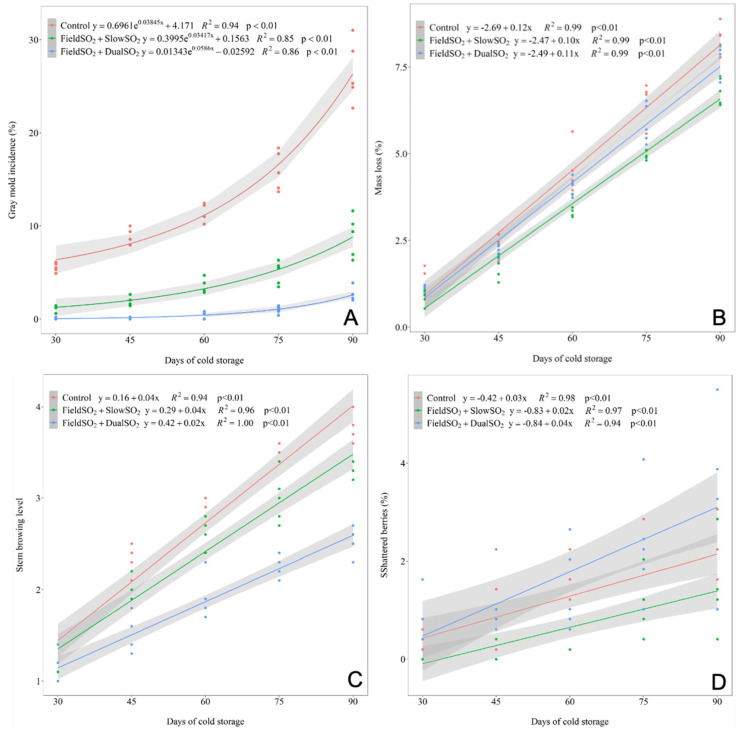
Regression analysis of the development of gray mold incidence (**A**), mass loss (**B**), stem browning (**C**), and shattered berries (**D**) of ‘Italia’ table grapes at 30, 45, 60, 75, and 90 days of cold storage (1.0 ± 1.0 °C), subjected to SO_2_-generating pads before packaging and during cold storage. Control: without SO_2_-generating pads, only a perforated plastic liner; FieldSO_2_: field ultra-fast SO_2_-generating pad before packaging; SlowSO_2_ and DualSO_2_: slow- and dual-SO_2_-generating pads, respectively, during cold storage—season of 2020.

**Figure 2 plants-13-02827-f002:**
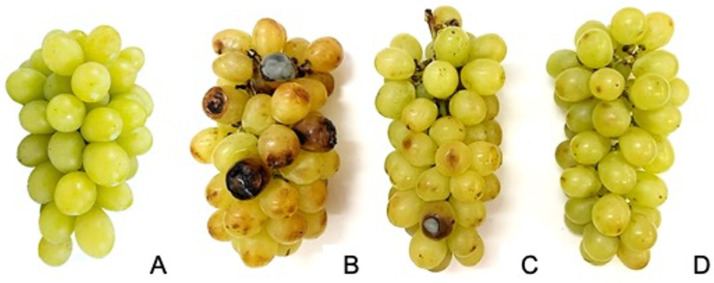
Bunches of ‘Italia’ table grapes before cold storage (**A**) and after 90 days of cold storage at 1.0 ± 1.0 °C, subjected to SO_2_-generating pads before packaging and during cold storage; control (**B**); field ultra-fast SO_2_-generating pad before packaging (FieldSO_2_) combined with slow-SO_2_-generating pad (SlowSO_2_) during cold storage (**C**); FieldSO_2_ combined with dual-SO_2_-generating pad (DualSO_2_) during cold storage (**D**).

**Figure 3 plants-13-02827-f003:**
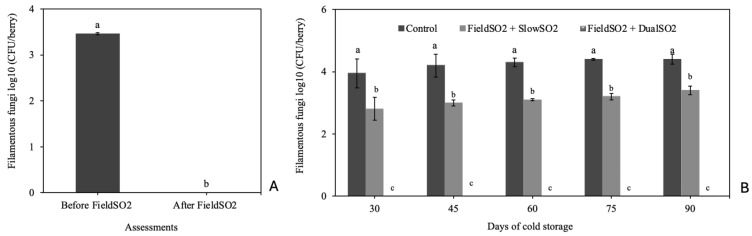
Population of filamentous fungi on berry skins of ‘Italia’ table grapes, before and after being subjected to the field ultra-fast SO_2_-generating pad (FieldSO_2_) before packaging (**A**), and at 30, 45, 60, 75 and 90 days of cold storage at 1.0 ± 1.0 °C subjected to the slow-SO_2_-generating (SlowSO_2_) or dual-SO_2_-generating (DualSO_2_) pads (**B**). Means followed by the same letters within columns are not different according to Tukey’s test (*p* < 0.05). CFU: colony-forming units—season of 2020.

**Figure 4 plants-13-02827-f004:**
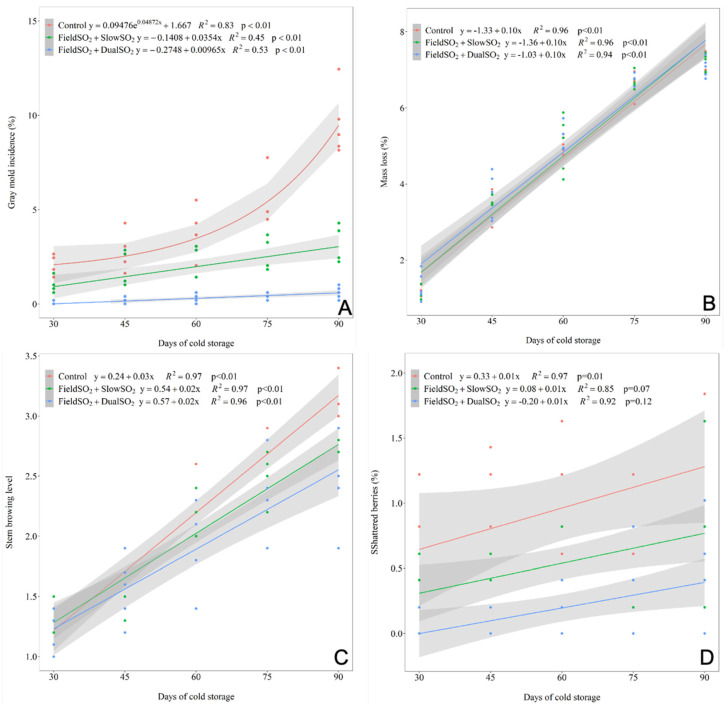
Regression analysis of the development of gray mold incidence (**A**), mass loss (**B**), stem browning (**C**), and shattered berries (**D**) of ‘Italia’ table grapes at 30, 45, 60, 75, and 90 days of cold storage (1.0 ± 1.0 °C), subjected to SO_2_-generating pads before packaging and during cold storage. Control: without SO_2_-generating pads, only a perforated plastic liner; FieldSO_2_: field ultra-fast SO_2_-generating pad before packaging; SlowSO_2_ and DualSO_2_: slow- and dual-SO_2_-generating pads, respectively, during cold storage—season of 2021.

**Figure 5 plants-13-02827-f005:**
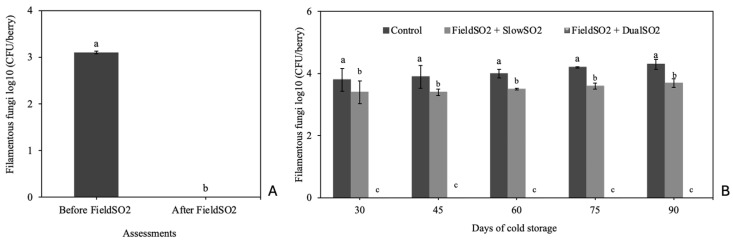
Population of filamentous fungi on berry skins of ‘Italia’ table grapes, before and after being subjected to the field ultra-fast SO_2_-generating pad (FieldSO_2_) before packaging (**A**), and at 30, 45, 60, 75 and 90 days of cold storage at 1.0 ± 1.0 °C subjected to slow-SO_2_-generating (SlowSO_2_) or dual-SO_2_-generating (DualSO_2_) pads (**B**). Means followed by the same letters within columns are not different according to Tukey’s test (*p* < 0.05). CFU: colony-forming units—season of 2021.

## Data Availability

Data are available upon request.

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
