# Peer review of "Association of SO_2_-Generating Pads before Packaging and during Cold Storage to Extend the Conservation of ‘Italia’ Table Grapes"

_plants, 2024, doi:10.3390/plants13192827_

Round 1
Reviewer 1 Report
Comments and Suggestions for Authors
The research paper titled "Association of SO2-generating Pads before Packaging and during Cold Storage to Extend the Conservation of ‘Italia' Table Grapes" aims to demonstrate the effectiveness of “FieldSO2,” an SO2-generating pad used in the field during harvest as a disinfection procedure. This study examines its use in combination with traditional slow and dual-release SO2-generating pads for controlling gray mold in table grapes. While similar studies have been conducted previously, this is the first focused specifically on the ‘Italia’ table grape cultivar under high decay pressure and extended storage periods.
The paper is well-written and organized, presenting results from two seasons with consistent data. However, several aspects need to be addressed before publication:
1. The authors should provide a more detailed explanation of the
differences among the SO2-generating pads used, particularly the specific
configurations that distinguish slow-release from fast-release pads.
2. The SO2 levels from “FieldSO2” were quantified using a dosimeter
(Gastec Passive Dosimeter Tube, Kanagawa, Japan), showing readings of 160, 410, and 600 ppm after 1, 2, and 3 hours, respectively. It should be clarified whether the evaluation was conducted using passive or active sampling methods, as diffusion-based measurement with a passive dosimeter requires a consideration of exposure time and is expressed in ppm-hours (ppm·h).
3. It is necessary to specify the ventilation area of the clamshells used in the experiments.
4. The authors should present the same information either in
figures or tables but not both. Therefore, they should select either Figure 1
or Table 1, and either Figure 2 or Table 2 to avoid redundancy.
5. In line 237, the authors are encouraged to discuss the commercial use of SO2 gassing chambers, as these may yield results similar to those produced by the proposed FieldSO2 technology.
Author Response
Dear Reviewer, first of all, thank you very much for your comments. We would like to inform that all of your suggestions were considered, and the proper changes were made. Below we reply your comments:
- The authors should provide a more detailed explanation of the
differences among the SO2-generating pads used, particularly the specific
configurations that distinguish slow-release from fast-release pads.
Answer: We add this information in the Material and Methods section.
2. The SO2 levels from “FieldSO2” were quantified using a dosimeter
(Gastec Passive Dosimeter Tube, Kanagawa, Japan), showing readings of 160, 410, and 600 ppm after 1, 2, and 3 hours, respectively. It should be clarified whether the evaluation was conducted using passive or active sampling methods, as diffusion-based measurement with a passive dosimeter requires a consideration of exposure time and is expressed in ppm-hours (ppm·h).
Answer: We clarified in the Material and Methods that we used the passive dosimeter to measure the SO2 released, and the unit was properly changed.
3. It is necessary to specify the ventilation area of the clamshells used in the experiments.
Answer: See in Material and Methods that we added this information (number and dimensions of the holes in the hinge).
4. The authors should present the same information either in
figures or tables but not both. Therefore, they should select either Figure 1
or Table 1, and either Figure 2 or Table 2 to avoid redundancy.
Answer: We selected Figures 1 and 2, and we removed Table 1 and Table 2.
5. In line 237, the authors are encouraged to discuss the commercial use of SO2 gassing chambers, as these may yield results similar to those produced by the proposed FieldSO2 technology.
Answer: We added a paragraph to discuss the commercioal use of SO2 gassing chamber in this section.
The authors.
Reviewer 2 Report
Comments and Suggestions for Authors
This MS studied the role of SO2 pads on grapes shelf life.
Below are the comments.
1. The phenotype of grapes during storage should be shown instead of the 90 days appearance.
2. 4.4 assessments, the subtitle is too brief and did not contain enough information.
3. For figure 3, what exactly are the filamentous fungi? How this experiment was done? Besides, I did not see any standard deviation.
Comments on the Quality of English Languagegood
Author Response
Dear Reviewer, first of all, thank you very much for your comments. We would like to inform that all of your suggestions were considered, and the proper changes were made. Below we reply your comments:
- The phenotype of grapes during storage should be shown instead of the 90 days appearance.
Answer: we added the phenotype of grape before the cold storage for comparions in Fig. 2. However, we don't have images from all assessments, only at 90 days of cold storage.
2. 4.4 assessments, the subtitle is too brief and did not contain enough information.
Answer: we changed the title by adding the full information about the assessments.
3. For figure 3, what exactly are the filamentous fungi? How this experiment was done? Besides, I did not see any standard deviation.
Answer: In Material and Methods we added this information (section 4.4).
The authors
Round 2
Reviewer 2 Report
Comments and Suggestions for Authors
the comments are resolved.